# Pathogenic Effects of Single or Mixed Infections of *Eimeria mitis*, *Eimeria necatrix,* and *Eimeria tenella* in Chickens

**DOI:** 10.3390/vetsci9120657

**Published:** 2022-11-24

**Authors:** Lixin Xu, Quanjia Xiang, Mongqi Li, Xiaoting Sun, Mingmin Lu, Ruofeng Yan, Xiaokai Song, Xiangrui Li

**Affiliations:** Ministry of Education (MOE) Joint International Research Laboratory of Animal Health and Food Safety, College of Veterinary Medicine, Nanjing Agricultural University, Nanjing 210095, China

**Keywords:** *Eimeria mitis*, *E. tenella*, *E. necatrix*, mixed infection

## Abstract

**Simple Summary:**

The objective of this study was to investigate the effects of the presence of *Eimeria mitis* on the outcome of *Eimeria tenella* or *Eimeria necatrix* experimental challenge infection and to demonstrate synergistic or antagonistic effects occurring among different species in mixed *Eimeria* infections. Co-infection of *E. mitis* and *E. tenella* led to lower body weight gain, severer lesions, and higher mortality of challenged birds compared to a single *E. tenella* infection. Moreover, although not statistically significant, there appears to be a reduction in body weight gain and an increase in oocyst shedding and mortality in *E. mitis*/*E. necatrix*-coinfected group compared to *E. necatrix*-infected group. These observations suggest that *E. mitis* could enhance disease mediated by *E. tenella*, whereas *E. mitis* does not seem to affect the virulence of *E. necatrix* but might also have a synergistic interaction with *E. necatrix* in infection. In conclusion, a synergistic relationship between *E. mitis* and *E. tenella*/*E. necatrix* was demonstrated using experimental co-infection models, thus suggesting that the common natural mixed infection of chicken coccidia in the field was probably the result of a synergistic effect of *Eimeria* spp. rather than an antagonistic one.

**Abstract:**

Avian *Eimeria* species vary in their replication location, fecundity, and pathogenicity. They are required to complete the development within the limited space of host intestines, and some synergistic or antagonistic effects occur among different *Eimeria* species. This study evaluated the impact of *Eimeria mitis* on the outcome of *Eimeria necatrix* or *Eimeria tenella* challenge infection. The severity of *E. mitis*/*E. necatrix* and *E. mitis*/*E. tenella* mixed infections were quantified by growth performance evaluation, survival rate analysis, lesion scoring, blood stool scoring, and oocyst output counting. The presence of *E. mitis* exacerbated the outcome of co-infection with *E. tenella*, causing high mortality, intestinal lesion score, and oocyst production. However, *E. mitis*/*E. tenella* co-infection had little impact on the body weight gain compared to individual *E. tenella* infection. In addition, the presence of *E. mitis* appeared not to enhance the pathogenicity of *E. necatrix*, although it tends to inhibit the growth of challenged birds and facilitate oocyst output and mortality in an *E. mitis*/*E. necatrix* co-infection model. Collectively, the results suggested a synergistic relationship between *E. mitis* and *E. tenella*/*E. necatrix* when sharing the same host. The presence of *E. mitis* contributed to disease pathology induced by *E. tenella* and might also advance the impact of *E. necatrix* in co-infections. These observations indicate the importance of accounting for differences in the relationships among different *Eimeria* species when using mixed infection models.

## 1. Introduction

Coccidiosis of the chicken is caused by species in the genus *Eimeria* and is one of the most frequent and severe enteric diseases for the poultry industry globally. Seven spe-cies of *Eimeria* are recognized in chicken, which are *Eimeria acervulina*, *Eimeria brunetti*, *Eimeria maxima*, *Eimeria mitis*, *Eimeria necatrix*, *Eimeria praecox*, and *Eimeria tenella*. In the intestine of birds, they exert robust site-specificity of development. For instance, *E. mitis* and *E. necatrix* have the same predilection sites: ileum and jejunum, whereas the parasitic region of *E. tenella* is focused in the caecum and duodenum [1]. The pathogenicity of different species varies. The most virulent species is *E. tenella,* which is responsible for important mortality. *E. necatrix* is highly virulent and causes a more chronic form of chicken coccidiosis [1,2,3]. *E. mitis* is generally thought to be the least virulent, and it does cause visible lesions [4].

Chicks are most susceptible to *E. mitis*, which affects their growth and nutrient absorption. Joyner found that the weight gains of chicks were evidently decreased when given above 5 × 10^5^ oocysts of *E. mitis*, and a number of deaths occurred when the birds were inoculated with 2.5 × 10^6^ oocysts [5]. Ruff and Edgar found that absorptions of L-methionine and glucose were significantly reduced in the intestine of infected broilers [6]. Fitz-Coy and Edgar suggested that *E. mitis* were pathogenic and suppressed the growth of broilers from day 3 post-inoculation [7]. In addition, infected birds produced watery droppings, and egg production was significantly reduced temporarily in hens [8].

In general, mixed infections are common in the field, and most commercial farms have the prevalence of coccidiosis caused by mixed *Eimeria* spp. populations [9,10,11,12,13]. Previous studies showed that the birds infected by two species of *E. acervuline*, *E. maxima*, and *E. brunetti* lost more weight than for a single infection. The output of oocysts was found to be related to the predilection sites in these species [14,15]. Ruff demonstrated that 5000 oocysts of *E. mitis* or 3000 mixed oocysts of equality *E. maxima*, *E. mitis*, and *E. acervuline* caused a significant decrease in growth rate and plasma pigment [16]. Additionally, Joyner and Norton revealed that mixed simultaneous infections of *E. mitis* and *E. acervuline* produced more severe effects on growth performance and lowered the ratio of villus height to total mucosal thickness in infected chicks [17]. However, single inoculation of *E. mitis* or *E. brunetti*, alongside a subsequent challenge with the other species, had no greater effects on growth performance than the single infection [17]. This observation raises the question of whether mixed infections of avian *Eimeria* species developing in different regions of the intestines produce a less or more severe pathology. Since *E. tenella* and *E. mitis* develop in different predilection sites and *E. mitis* and *E. necatrix* develop in similar sites, we investigated the pathogenetic effects of mixed infections of *E. mitis*, *E. necatrix*, and *E. tenella* in the hosts. The co-infections of *E. tenella* with *E. mitis* produced a more significant pathology in challenged birds than single *E. tenella* infection, whereas the severity of pathology induced by exposure to mixed *E. mitis* and *E. necatrix* challenge was not significantly augmented compared to a single *E. necatrix* challenge.

## 2. Materials and Methods

### 2.1. Animal and Parasite

One-day-old yellow broilers were obtained from a commercial hatchery and reared in coccidian-free conditions. The broilers were fed with a standard ration *ad libitum*. On Day 0 (the day of inoculation), the chickens were 15 days old.

### 2.2. Experiments

The chickens (15-day-old) were randomized into six groups balanced by weight. Each group contained 30 chickens and was divided into three sub-groups. On Day 0, the birds were inoculated with varying combinations of *Eimeria* oocysts, and the doses administered to the birds are listed (Table 1).

The chickens were observed daily after infection for bloody stools, morbidity, and mortality. Feces were collected on day 4 post-infection until the end of the trial. All chickens were humanely sacrificed by cervical dislocation and dissected on day 8 post-infection and scored for weight gain, the output of oocyst, and intestinal lesions throughout the trial. The rate of body weight gain was calculated by denoting weight gains of challenged birds as a percentage of uninfected controls. The survival proportions of each *Eimeria*-challenged group were expressed as the percentage of surviving chickens at the end of the test compared with the initial number of chickens. The fecal score was calculated by evaluating the appearance of the fecal droppings from Day 4 to Day 7 based on the criteria suggested by Morehouse and Baron [18]. The gradation of bloody stools for each group was defined using a 4-point scoring system [18]. Lesion scoring was carried out referring to the criteria of Johnson and Reid [19]. Briefly, small intestine and cecal tissue samples were collected *post-mortem*, and the severity of lesions within the gut was determined using the widely accepted 0 to 4 scoring system.

For each group, 24-h-collections of feces were carried out to determine oocyst output using a sampling regime described by Long et al. [20]. Moreover, *Eimeria* species were differentiated based on oocyst size, the smaller of the two oocysts being *E. mitis*. In addition, the presence of *E. tenella*, *E. mitis*, and *E. necatrix* oocysts was confirmed by internal transcribed spacer 1 (ITS-1) PCR methods using previously validated primers [21].

From Day 4 to Day 8, total oocysts of feces counts were made. For intestinal oocyst counting, the cecum was sampled from the birds challenged with *E. tenella* after the chickens were killed. In addition, the segment of the middle third (about 10 cm before and after the yolk pedicle) of the small intestine was taken for *E. necatrix* oocyst counting. Finally, the whole intestinal tract of infected chickens was taken for *E. mitis* oocyst counting. The oocysts output is the sum of oocysts in the feces and those in the intestine.

### 2.3. Statistical Analysis

Analysis of variance (ANOVA) test supplemented with GraphPad Prism 7.04 was conducted to assess the significant difference in weight gains, blood stool score, survival proportions, and lesion scores among each group. In addition, an unpaired *t*-test was carried out to determine the statistically significant difference in oocyst output across varying experimental groups.

## 3. Results

### 3.1. Growth Rate in Each Eimeria-Challenged Group

Mean weight gain of birds after inoculation with pure or mixed oocysts of *E. tenella*, *E. mitis*, and *E. necatrix* are shown in Table 1. The average weight gain of the control group was significantly higher than *E. tenella*-infected groups (*p* < 0.05), whereas there was no statistical significance in average weight gain between *E. mitis*-infected/*E. necatrix*-infected and control groups (*p* > 0.05) (Table 1). Meanwhile, *E. mitis* and *E. tenella* co-infection resulted in a notably reduced growth rate of birds compared to a PBS mock infection (*p* < 0.05). In contrast, *E. mitis* and *E. necatrix* co-infection did not exert significant impacts on the weight gain of challenged birds compared to the PBS mock infection (*p* > 0.05). The reduction rate of the mean body weight gain in *E. tenella*-infected and *E. mitis*/*E. tenella* co-infected groups were 61.6% and 60.0%, respectively (Table 1). Notably, the incorporation of *E. mitis* to *E. tenella* or *E. necatrix* in the inoculum, although increasing the inoculum dose of *Eimeria* parasites, did not significantly alter the average weight gain compared to individual *E. tenella* or *E. necatrix* infection (*p* > 0.05) (Table 1).
vetsci-09-00657-t001_Table 1Table 1The effect of *E. mitis* and *E. tenella*, *E. necatrix* in pure and mixed infections in chicks.GroupNo. of Oocysts Inoculated on Day 0 (×10^4^)Mean Weight Gain (±SE)Rate of Average Weight Gain (%)Survival Rate (%)Control1 mL PBS62.92 ± 4.04 ^a^100100*E. mitis*10 *E. mitis*50.44 ± 3.33 ^ab^80.2100*E. mitis* and *E. tenella*10 *E. mitis* + 5 *E. tenella*38.72 ± 3.61 ^bc^61.623.3*E. tenella*5 *E. tenella*37.27 ± 3.04 ^c^60.090*E. mitis* and *E. necatrix*10 *E. mitis* + 10 *E. necatrix*50.29 ± 3.07 ^ab^80.093.3*E. necatrix*10 *E. necatrix*53.11 ± 2.48 ^ad^84.4100Values in each column followed by a different letter are significantly different, *p* < 0.05.


### 3.2. Survival Proportion

The survival proportions of all groups are presented in Table 1. The survival proportion of the *E. mitis*/*E. tenella* mixed infection group was only 23.3%, remarkably decreased than the *E. tenella* infection group (90%). Meanwhile, the survival proportion of *E. mitis*/*E. tenella* co-infection group also differed significantly from other groups (Table 1). In addition, there was 6.7% mortality occurred in *E. mitis*/*E. necatrix* mixed infection group (Table 1). There were no deaths in the single *E. mitis* or *E. necatrix* infection group. The results indicated that *E. mitis* exacerbated the pathogenic abilities of *E. tenella* and might have a synergistic effect with *E. tenella* for pathogenicity. However, *E. mitis* appeared not significantly to exacerbate the outcome of co-infection with *E. necatrix*, although lower survival proportion of *E. mitis*/*E. necatrix* co-infection group was observed but reaching no significance.

### 3.3. Lesion Score

As shown in Figure 1, either single or mixed infections of three *Eimeria* spp. caused apparent lesions in the intestinal tract, indicating that the infectious doses used in this study were sufficient to induce pathogenic effects in the hosts. The lesion score in *E. mitis*/*E. tenella* mixed infection group reached 3.867, suggesting tremendous intestinal damage caused by the challenge of these two species. Besides, the lesion caused by *E. mitis*/*E. tenella* co-infection was more severe than a single *E. tenella* infection (the lesion score was 1.733) (*p* < 0.05) (Figure 1), indicating the apparent positive correlation between *E. mitis* and *E. tenella* in inducing intestinal damage. On the contrary, mixed infection with *E. mitis* and *E. necatrix* did not induce more severe intestinal lesions than a single *E. necatrix* infection (*p* > 0.05) (Figure 1). Moreover, the lesion scores in *E. mitis*/*E. necatrix* co-infection group appears to be lower than *E. necatrix* infection group (lesion score 2.067 vs. 2.233), although not statistically significant. Thus, it was likely that there was not enough space for the parasitism infection of *E. mitis* and *E. necatrix*, given that both *Eimeria* species develop in host small intestines.

### 3.4. Blood Stool Score

Bloody stools were observed in the single *E. tenella* or *E. necatrix* infection, *E. mitis*/*E. necatrix* co-infection, and *E. mitis*/*E. tenella* co-infection groups, respectively (Figure 2). The mean blood stool scores in these groups were 1.7, 0.033, 1.733, and 0.1, respectively. In contrast, no bloody stools were noted with pure *E. mitis* infection (Figure 2), which is consistent with the low virulence of *E. mitis*.

### 3.5. The Oocyst Output

The presence of *E. necatrix*, *E. tenella*, and *E. mitis* oocysts in the feces of varying challenged groups were validated by preferentially amplifying the internal transcribed spacer 1 (ITS-1) regions. The PCR assays detected *the E. mitis* population in the *E. mitis* infection group (Figure 3a, lane 1; approximately 306 bp), mixed *E. mitis*/*E. tenella* infection group (Figure 3a, lane 2; approximately 306 bp), and *E. mitis*/*E. necatrix* co-infection group (Figure 3a, lane 4; approximately 306 bp). In addition, the *E. tenella* population was only identified in *E. tenella*-infection group (Figure 3a, lane 2; approximately 278 bp) and mixed *E. mitis*/*E. tenella* infection group (Figure 3a, lane 3; approximately 278 bp). Moreover, *E. necatrix* population was solely found in the *E. necatrix* infection group (Figure 3a, lane 4; approximately 383 bp) and *E. mitis*/*E. necatrix* co-infection group (Figure 3a, lane 5; approximately 383 bp). The oocyst output of all groups was shown as a histogram in Figure 3b. The output of *E. mitis* oocysts was significantly reduced in *E. mitis*/*E. tenella* co-infection group (*p* < 0.05) (Figure 3b). In addition, although not statistically significant, the output of *E. tenella* oocysts in *E. mitis*/*E. tenella* co-infection group appeared to be decreased compared to the *E. tenella* infection group (Figure 3b). All these observations indicated that mixed infections might have an impact on the reproduction of each *Eimeria* spp. In the *E. mitis*/*E. necatrix* co-infection group, although reaching no statistical significance, the oocyst output of *E. mitis* decreased compared to a single *E. mitis* infection, whereas that of *E. necatrix* increased compared to a single *E. necatrix* infection (Figure 3b). It is likely that *E. necatrix* was dominant over *E. mitis* in the case of co-infection of *E. mitis* and *E. necatrix*.

## 4. Discussion

Avian coccidiosis is a devasting enteric disease with a worldwide prevalence. Currently, the main strategy to combat coccidiosis is still the usage of chemical drugs. However, chemical drugs are prone to suffer drug resistance and drug residues. With increasing concerns about drug residues and stringent requirements by governments and civil agencies for the use of chemical drugs, immunization by vaccines to elicit substantial protective immunity is a superb alternative strategy. However, the immune protection provided by vaccines varies depending on chicken breed, dosing schedule, and infecting *Eimeria* species. Different *Eimeria* species require the use of live vaccines specific for target species, and in some cases, no or little protective immunity was induced in the hosts challenged with a heterologous species. In addition, different *Eimeria* species vary in their replication location, fecundity, and pathogenicity. They are required to complete the development within the limited space of the host intestine. Therefore, there could be some synergistic or antagonistic relationships among heterologous *Eimeria* species. There were reports of *E. mitis* content increased from 30% to 98% in a mixed sample containing *E. acervuline*, *E. necatrix*, *E. tenella*, *E. mitis*, and *E. maxima* following the use of *E. mitis*-free live vaccines [22]. This suggested that the presence of other *Eimeria* may inhibit *E. mitis* development. Further investigation is warranted to determine the relationship among these *Eimeria* species when they share the same host.

Tyzzer, who found and named *E. mitis*, revealed that the effects of *E. mitis* infection were not manifest in healthy chickens. *E. mitis* caused minor damage because of its scattered distribution in the small intestine. Even the repeated and constant infection with a hefty dose of *E. mitis* oocysts did not affect the activity of challenged birds and did not induce apparent lesions at necropsy [4]. However, a subsequent study demonstrated that *E. mitis* infection could cause poor growth performance including low body weight gain [5], nutrient malabsorption [6] and reduced feed conversion, and clinical symptoms including spale skin pigmentation and a drain-like appearance of feces [7]. Of note, *E. mitis* can enhance the pathogenicity of *E. acervuline* in chickens chronically afflicted with *Plasmodium juxtanucleare* [23]. Meanwhile, the co-infection of *E. mitis* and low-pathogenicity reoviruses can adversely affect the growth performance of infected birds [16,24].

Epidemiological studies demonstrated that mixed *Eimeria* infections occurred naturally and were frequently found in poultry. In natural infections, chickens were infected with mixed *Eimeria* oocysts containing at least two species and sometimes more than six species [9,25,26]. The epidemiological surveys revealed that the rates of *E. mitis*-containing infections are inconsistent, with some as high as 96% [27]. The effects of *E. mitis* on other *Eimeria* species are not well studied [17,28]. Different avian *Eimeria* species have traditionally been considered to have specific sites of parasitism in the hosts [29]. However, the specific sites can be altered by factors such as the immunosuppressive drugs that the host is receiving. Thus, avian *Eimeria* species are not permanently restricted to site-specificity of development [30], and there may also be crowding effects that alter the parasitized sites [31]. Multiple *Eimeria* species simultaneously parasitize the chicken intestinal tract to form a relationship with each other. Joyner and Norton suggested that a synergistic effect occurred if the two *Eimeria* spp. did not parasitize the same intestinal segment, and the hazard effect of their mixed infection was greater than that of individual infection [17]. In this study, the mixed infection of *E. tenella* and *E. mitis* induced a higher pathology severity than a single *E. tenella* challenge infection, consistent with Joyner and Norton’s observation. Herein, co-infection of *E. tenella* and *E. mitis* induced significant changes in afflicted birds, and the survival proportion was significantly reduced from 90% to 23.3%. The most remarkable change detected in challenged birds was severe caecal pathology. In addition, the lesion score in *E. mitis*/*E. tenella* co-infection group reached 3.867, which may contribute to the high mortality rate of challenged birds.

Two species parasitizing at the same site may have competitive relationships, as exemplified by *E. mitis* and *E. brunetti* [32]. In this study, although not statistically significant, we identified a decrease in *E. mitis* oocyst production and an increase in *E. necatrix* oocysts in *E. mitis*/*E. necatrix* mixed infection group when compared to single *E. mitis* or *E. necatrix* infection. Thus, in the case of *E. mitis*/*E. necatrix* co-infection, *E. mitis* was likely not dominant over *E. necatrix*, and we anticipated that there could be some antagonistic relationships between *E. mitis* and *E. necatrix*. However, based on the observations of decreased weight gains and elevated blood stool score and mortality in *E. mitis*/*E. necatrix* mixed infection group (reaching no significance), it was likely that there was also a synergistic interaction between *E. mitis* and *E. necatrix* when sharing the site of infection within the host gut. Given that the results of the survival rates, lesion score, and blood stool score between *E. mitis*/*E. necatrix* mixed infection group and single *E. mitis* or *E. necatrix* infection were inconsistent, it would therefore seem prudent to conclude that we still do not yet fully understand the exact relationship between these two *Eimeria* species. Future efforts are needed to make it clear.

## 5. Conclusions

In an *E. mitis*/*E. tenella* co-infection model, the presence of *E. mitis* exacerbated the outcome of co-infection with *E. tenella*, causing elevated intestinal lesions, oocyst production, and mortality. In addition, mixed infection of these two *Eimeria* species had negligible impacts on the weight gain of challenged birds compared with individual *E. tenella* infection. Thus, the results support the view that a synergistic effect occurred between *E. mitis* and *E. tenella* when sharing the same host, and the presence of *E. mitis* contributes to disease pathology. However, in the case of *E. mitis*/*E. necatrix* co-infection, *E. mitis* does not seem to significantly affect the pathogenicity of *E. necatrix,* although there appeared to be a synergistic interaction between *E. mitis* and *E. necatrix*. Collectively, we provided some new information on the effect that *E. mitis* has on *E. tenella* or *E. necatrix* infection, and the results indicate the importance of accounting for differences in the relationships among different *Eimeria* species when using mixed infection models.

## Figures and Tables

**Figure 1 vetsci-09-00657-f001:**
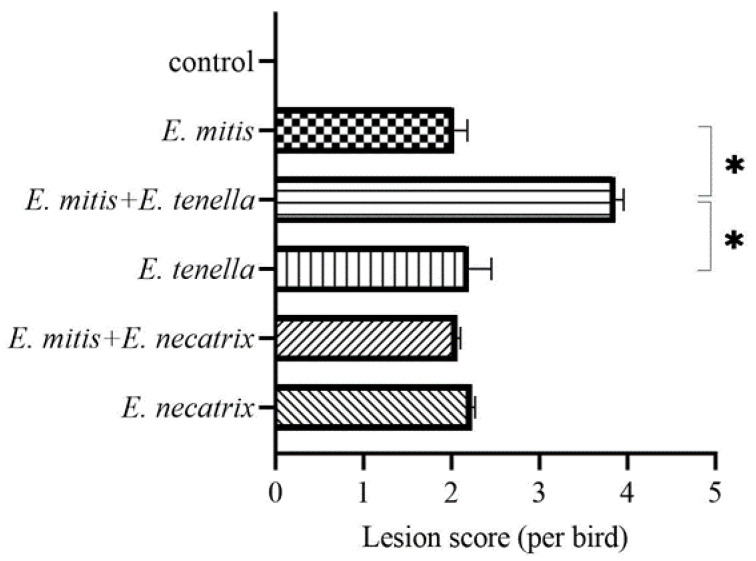
The lesion score of chicken intestines with single and mixed infections of *Eimeria* species. The asterisk denotes that two groups significantly differed (*p* < 0.05).

**Figure 2 vetsci-09-00657-f002:**
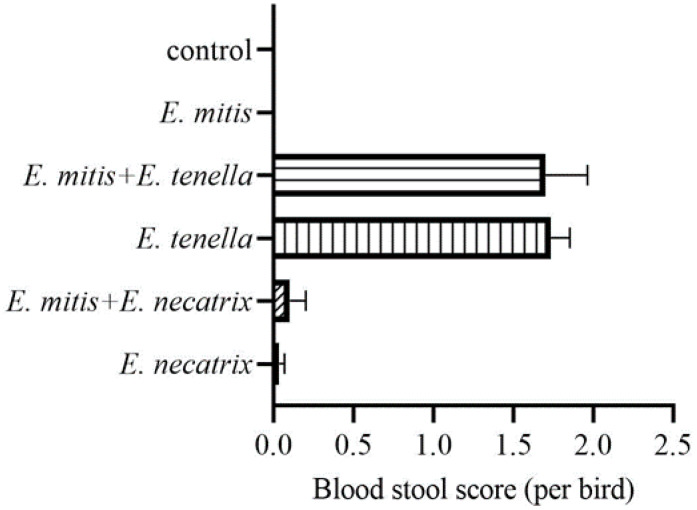
The blood stool score of chickens with single and mixed infections of *Eimeria* species.

**Figure 3 vetsci-09-00657-f003:**
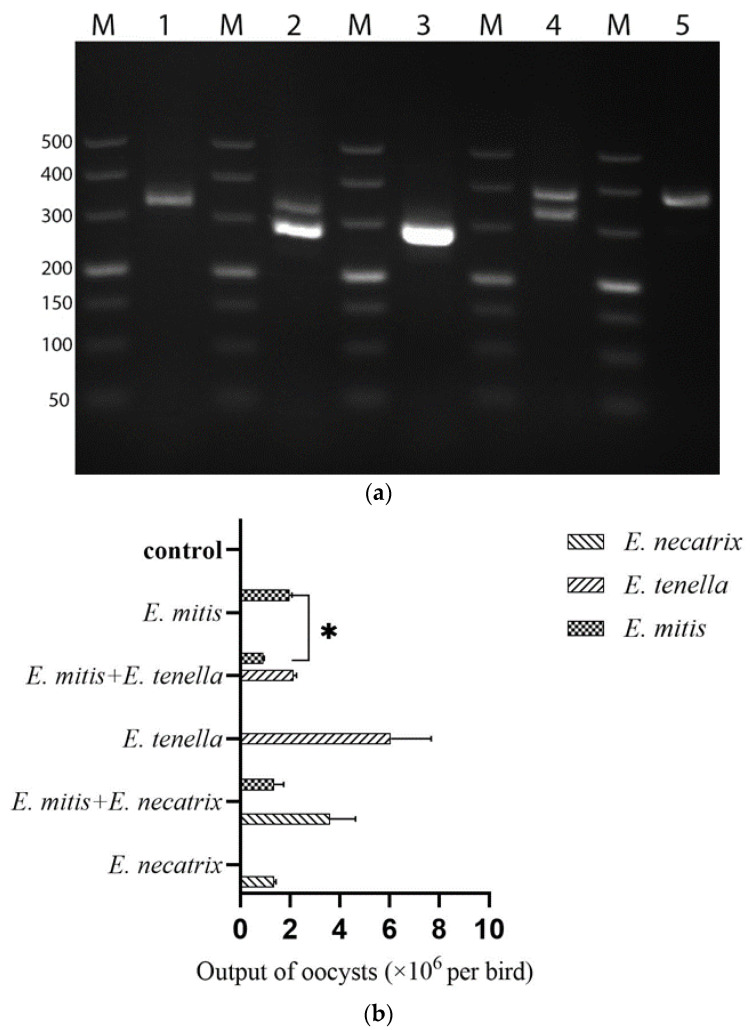
Oocyst shedding in the feces using single and mixed infections of each species. (**a**) Species-specific amplification by PCR assays. Lane M: molecular weight ladder; Lane 1: *E. mitis*; Lane 2: *E. mitis*/*E. tenella*/; Lane 3: *E. tenella*; Lane 4: *E. mitis*/*E. necatrix*; Lane 5: *E. necatrix*. (**b**) Comparison of oocyst output in single and mixed infection groups. Two groups that differ significantly are marked by a capped line (* *p* < 0.05).

## Data Availability

The data presented in this study are available on request from the corresponding author.

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
