# Peer review of "Pathogenic Effects of Single or Mixed Infections of Eimeria mitis, Eimeria necatrix, and Eimeria tenella in Chickens"

_vetsci, 2022, doi:10.3390/vetsci9120657_

Round 1

Reviewer 1 Report

Title – adapt to read as:  Pathogenic effects of single or mixed infections of Eimeria mitis, Eimeria necatrix and Eimeria tenella in chickens

Affiliations – explain the meaning o MOE or delete it

Simple summary – please clearly mention whether infections were natural or experimental

Write out every species name the first time it used, i.e. Eimeria mitis, etc.

Pathogenicity is a qualitative concept – every time the authors are referring to quantitative effects they should use “virulence”

Line 10 – adapt: … investigate EFFECTS OF the presence

Line 16 – suggest (instead of suggested)

If there were “no significant differences in body weight gain, gut lesions, and mortality between E. mitis/E. necatrix-coinfected group and E. necatrix-infected group”, the authors cannot conclude that the pathogenicity/virulence of E. necatrix might be impeded by an antagonistic relationship with E. mitis”.

What the authos should conclude is that E. mitis does not seem to affect the pathogeneiity/virulence of E. necatrix – this should be corrected in the simple summary, abstract and main text

Line 20, etc. – write Eimeria spp. (instead of Eimeria spp.)

Line 29 – replace “weight gain than” with “compared to”

Line 42 – is there any criteria or displaying the species of Eimeria like this? Would not alphabetical order be more adequate?

Line 46 – most pathogenic – this is not a correct phrasing – replace with “most virulent”

Line 47 – replace significant with important

Highly pathogenic should be “highly virulent”

Line 49 – least virulent

Line 50 – adapt: Chicks ARE most susceptible to E. mitis, WHICH AFFECTS THEIR growth and nutrient absorption

Line 51 – write “chicks” (insted of the chicks)

Joyner should be Joyner and Norton – change accordingly throughout the manuscript

Lines 55 and 57 – broilers (without the) and hens (also without the), respectively

Line 58 – are common (instead of were common)

English should be reviewed by a native speaker of the language – this reviewer is not poiting out all the aspects that need to be corrected

Line 60 – Eimeria acervulina

Line 61 – than FOR a single infection

Line 62 – Ruff demonstrated…

Line 66 – Joyner and Norton revelead…

Line 82 – 15 days old

Line 84 – 15-day-old

Line 85 – which criteria have determined this sample size, i.e. 30 chickens?

Line 93 – survival rate should be “survival proportion” – change accordingly throughout the manuscript

Lines 95-98 – text is not clear – please rewrite

Line 101 – smaller (instead of lesser)

Statistical analysis – not clear for what purpose ANOVA was used

Were data normally distributed?

Line 126, etc. – one decimal place would be enough when presenting percentages

Line 133 – ARE presented

Line 136 – than IN other groups

Line 137 – write Latin names in italics

Not clear: In addition, fatalities (6.7% mortality) occurred after E. mitis was mixed with E. necatrix oocyst (Table 1).

What do the authors mean by fatality? Please standardize definitions

Line 146 – what is the meaning of “tremendous”?

Line 163 – low virulence

Line 195 – replace develop with “suffer”

References – they are not presented homogeneously – some titles have uppercase, while others have lowercase; some jornal titles are written out, while others are abbreviated; some have DOI, while other do not; etc.

Author Response

Title – adapt to read as:  Pathogenic effects of single or mixed infections of Eimeria mitis, Eimeria necatrix and Eimeria tenella in chickens

Response: Revised accordingly. Thanks.

Affiliations – explain the meaning o MOE or delete it

Response: Revised.

Simple summary – please clearly mention whether infections were natural or experimental

Response: Specified accordingly.

Write out every species name the first time it used, i.e. Eimeria mitis, etc.

Response: Revised throughout the MS.

Pathogenicity is a qualitative concept – every time the authors are referring to quantitative effects they should use “virulence”

Response: Thanks for the suggestion. Revised accordingly.

Line 10 – adapt: … investigate EFFECTS OF the presence

Response: Revised (Line 10).

Line 16 – suggest (instead of suggested)

Response: Revised (Line 17).

If there were “no significant differences in body weight gain, gut lesions, and mortality between E. mitis/E. necatrix-coinfected group and E. necatrix-infected group”, the authors cannot conclude that the pathogenicity/virulence of E. necatrix might be impeded by an antagonistic relationship with E. mitis”.

What the authos should conclude is that E. mitis does not seem to affect the pathogeneiity/virulence of E. necatrix – this should be corrected in the simple summary, abstract and main text

Response: Thanks for your suggestions. We have thoroughly revised the conclusions throughout the MS.

Line 20, etc. – write Eimeria spp. (instead of Eimeria spp.)

Response: Revised (Line 21).

Line 29 – replace “weight gain than” with “compared to”

Response: Revised (Line 31).

Line 42 – is there any criteria or displaying the species of Eimeria like this? Would not alphabetical order be more adequate?

Response: As far as we know, there should not be strict criteria for displaying the species of Eimeria, although that is a constructive suggestion. It was revised accordingly (Lines 44-45).

Line 46 – most pathogenic – this is not a correct phrasing – replace with “most virulent”

Response: Revised (Line 49).

Line 47 – replace significant with important

Response: Revised (Line 49-50).

Highly pathogenic should be “highly virulent”

Response: Revised (Line 50).

Line 49 – least virulent

Response: Revised (Line 51).

Line 50 – adapt: Chicks ARE most susceptible to E. mitis, WHICH AFFECTS THEIR growth and nutrient absorption

Response: Revised (Line 53).

Line 51 – write “chicks” (insted of the chicks)

Response: Revised (Line 53).

Joyner should be Joyner and Norton – change accordingly throughout the manuscript

Response: Revised throughout the MS.

Lines 55 and 57 – broilers (without the) and hens (also without the), respectively

Response: Revised accordingly (Lines 59 and 60).

Line 58 – are common (instead of were common)

Response: Revised accordingly (Line 61).

English should be reviewed by a native speaker of the language – this reviewer is not poiting out all the aspects that need to be corrected

Response: We have asked a native speaker for language editing as requested. Appreciate your help as well.

Line 60 – Eimeria acervuline

Response: Revised accordingly (Line 63).

Line 61 – than FOR a single infection

Response: Revised (Line 64).

Line 62 – Ruff demonstrated…

Response: Revised (Line 65).

Line 66 – Joyner and Norton revelead…

Response: Revised (Line 68).

Line 82 – 15 days old

Response: Revised (Line 85).

Line 84 – 15-day-old

Response: Revised (Line 87).

Line 85 – which criteria have determined this sample size, i.e. 30 chickens?

Response: The sample size was determined based on the suggestions by Francesca et al. (2021) for carrying out studies on the immunology and pathogenesis of Eimeria infection.

Line 93 – survival rate should be “survival proportion” – change accordingly throughout the manuscript

Response: Revised throughout the MS.

Lines 95-98 – text is not clear – please rewrite

Response: Revised accordingly (Lines 98-104).

Line 101 – smaller (instead of lesser)

Response: Revised (Line 107).

Statistical analysis – not clear for what purpose ANOVA was used

Were data normally distributed?

Response: Specied (Lines 117-121). The data were not normally distributed.

Line 126, etc. – one decimal place would be enough when presenting percentages

Response: Revised throughout the MS.

Line 133 – ARE presented

Response: Revised (Line 140).

Line 136 – than IN other groups

Response: Revised (Line 143).

Line 137 – write Latin names in italics

Response: Revised (Line 144).

Not clear: In addition, fatalities (6.7% mortality) occurred after E. mitis was mixed with E. necatrix oocyst (Table 1).

What do the authors mean by fatality? Please standardize definitions

Response: Rephrase the sentence (Lines 144-145).

Line 146 – what is the meaning of “tremendous”?

Response: massive.

Line 163 – low virulence

Response: Revised (Line 174).

Line 195 – replace develop with “suffer”

Response: Revised (Line 210).

References – they are not presented homogeneously – some titles have uppercase, while others have lowercase; some jornal titles are written out, while others are abbreviated; some have DOI, while other do not; etc.

Response: The reference section has been reformatted. Thanks for your valuable comments.

Reviewer 2 Report

The manuscript entitles “Pathogenic Effects of Single or Mixed Infections of Eimeria mitis, E. tenella and E. necatrix in Chickens” presents a study to investigate the presence of E. mitis on the outcome of E. tenella or E. necatrix challenge infection. According to the authors, E. mitis could enhance disease pathology mediated by E. tenella, whereas the pathogenicity of E. necatrix might be impeded by an antagonistic relationship with E. mitis. Although the authors did not conduct in-depth mechanism research, I think the manuscript provides a reference to strategy application of the vaccine immunization in avian coccidiosis.

In addition, I think there are some questions that need to be raised. 

1  Does this conclusion need to consider the immune status of the host (chicken)?

2  Line 118-119,  the average weight gain of E. necatrix-infected groups was not significantly lower than the control group (P in not  < 0.05).

3 Line 140.  E. mitis appeared not significantly to exacerbate the outcome of co-infection with E. necatrix? However, we can see a decline in both weight gain and survival rate in E. mitis and E. tenella mixed infection group. 

4 The asterisk in the figure was not noted. 

5 The section "3.5. The oocyst output" is very confusing. Please reorganize the text revision.

Author Response

The manuscript entitles “Pathogenic Effects of Single or Mixed Infections of Eimeria mitis, E. tenella and E. necatrix in Chickens” presents a study to investigate the presence of E. mitis on the outcome of E. tenella or E. necatrix challenge infection. According to the authors, E. mitis could enhance disease pathology mediated by E. tenella, whereas the pathogenicity of E. necatrix might be impeded by an antagonistic relationship with E. mitis. Although the authors did not conduct in-depth mechanism research, I think the manuscript provides a reference to strategy application of the vaccine immunization in avian coccidiosis.

In addition, I think there are some questions that need to be raised.

1  Does this conclusion need to consider the immune status of the host (chicken)?

Response: That is an excellent question. Well, if the birds are under immunosuppression (i.e., IBV vaccination and stress), the conclusion could be completely different. In this study, we used healthy birds to carry out challenge infection studies and it should be ok to conclude such as relationship between E. mitis and E. tenella/E. necatrix in co-infections. Thanks anyway for providing another fundamental perspective.

2  Line 118-119,  the average weight gain of E. necatrix-infected groups was not significantly lower than the control group (P in not  < 0.05).

Response: Revised accordingly (Lines 125-126).

3 Line 140.  E. mitis appeared not significantly to exacerbate the outcome of co-infection with E. necatrix? However, we can see a decline in both weight gain and survival rate in E. mitis and E. tenella mixed infection group.

Response: Yes, there appeared to be a decline in both weight gain and survival rate in E. mitis/E. necatrix mixed infection group compared to the E. necatrix infection group, although it is not statistically significant. Our previous conclusion on the antagonistic relationship between E. mitis and E. necatrix was made based on the observation of lesions in the E. necatrix infection group and E. mitis/E. necatrix mixed infection group. However, this could not be the case, given the results of weight gain, survival rate, and oocyst output. As a result, we corrected our conclusion and suggested a synergistic relationship between E. mitis and E. necatrix as well. Thanks for your insightful comment.

4 The asterisk in the figure was not noted.

Response: Specified in Line 168. Thanks.

5 The section "3.5. The oocyst output" is very confusing. Please reorganize the text revision.

Response: Reorganized accordingly. Thanks a lot for your valuable and constructive comments.

Reviewer 3 Report

This article submitted by Xu at al describes the pathogenic effect of the mixed infections of E. mitis, E. tenella and E. necatrix accompanied by the control (mock infection). The article is good written, and the scientific content is optimal for the journal. I would recommend improving the description of the methods including euthanasia process, the description of certain parameters to grade the intestinal lesions, and indicating the IACUC permits to handled chickens.  

Line 16: delete “pathology”

Line 17: delete “To conclude, in this study,” and write “In conclusion, a synergistic…”

Line 50: Delete “The”

Line 59-61: the sentence is not clear. Try to change the word “gave” by “infected by”

Line 90: what do you mean with “humanely sacrificed”? be more specific.

Line 95-96: the sentence is confused. How can the blood stools be scored the day before the birds were born (birth)?

Line 97: even though the authors cite Johnson and Reid for lesion scores, it would be better to add two or three sentences summarizing the criteria for the lesion scores.

Line 101: consider using “smaller” than “lesser”

Line 140: “disease pathogenesis” is not accurate for this sentence, if you mean “…for pathogenicity.”?

Line 223: since Eimeria are internal coccidia, it should be considered as infected and not infested.

Line 252: spices?

Author Response

This article submitted by Xu at al describes the pathogenic effect of the mixed infections of E. mitis, E. tenella and E. necatrix accompanied by the control (mock infection). The article is good written, and the scientific content is optimal for the journal. I would recommend improving the description of the methods including euthanasia process, the description of certain parameters to grade the intestinal lesions, and indicating the IACUC permits to handled chickens. 

Response: Thanks for your constructive suggestions. We’ve thoroughly revised the MS based on your valuable comments. Particularly, relevant information regarding the permission from the IACUC of Nanjing Agriculture University (Lines 293-296), how intestinal lesions were determined (Lines 101-104), and how birds were sacrificed (Line 93) were provided in the MS.

Line 16: delete “pathology”

Response: Revised (Line 17).

Line 17: delete “To conclude, in this study,” and write “In conclusion, a synergistic…”

Response: Revised (Lines 18-19 ).

Line 50: Delete “The”

Response: Revised (Line 53).

Line 59-61: the sentence is not clear. Try to change the word “gave” by “infected by”

Response: Revised accordingly (Line 63). Thanks.

Line 90: what do you mean with “humanely sacrificed”? be more specific.

Response: Birds were sacrificed by cervical dislocation. Revised accordingly (Line 93).

Line 95-96: the sentence is confused. How can the blood stools be scored the day before the birds were born (birth)?

Response: We have thoroughly revised the section in which the fecal scoring was carried out (Lines 98-101).

Line 97: even though the authors cite Johnson and Reid for lesion scores, it would be better to add two or three sentences summarizing the criteria for the lesion scores.

Response: Thanks for your suggestion. We have provided more details with regard to how the lesion scores were determined (Lines 101-104).

Line 101: consider using “smaller” than “lesser”

Response: Corrected (Line 107). Thanks.

Line 140: “disease pathogenesis” is not accurate for this sentence, if you mean “…for pathogenicity.”?

Response: Revised accordingly (Line 147). Thanks.

Line 223: since Eimeria are internal coccidia, it should be considered as infected and not infested.

Response: Revised (Line 238).

Line 252: spices?

Response: Revised accordingly (Line 271). Thanks again for your critical review and valuable comments.

Round 2

Reviewer 1 Report

The authors have addressed all of my comments and followed almost all of my suggestions.